# UniISP: A Unified ISP Framework for Both Human and Machine Vision

## Abstract

Compared to RGB images, raw sensor data provides a richer representation of information, which is crucial for accurate recognition, particularly under challenging conditions such as low-light environments. The traditional Image Signal Processing (ISP) pipeline generates visually pleasing RGB images for human perception through a series of steps, but some of these operations may adversely impact the information integrity by introducing compression and loss. Furthermore, in computer vision tasks that directly utilize raw camera data, most existing methods integrate minimal ISP processing with downstream networks, yet the resulting images are often difficult to visualize or do not align with human aesthetic preferences. This paper proposes UniISP, a novel ISP framework designed to simultaneously meet the requirements of both human visual perception and computer vision applications. By incorporating a carefully designed Hybrid Attention Module (HAM) and employing supervised learning, the proposed method ensures that the generated images are visually appealing. Additionally, a Feature Adapter module is introduced to effectively propagate informative features from the ISP stage to subsequent downstream networks. Extensive experiments demonstrate that our approach achieves state-of-the-art performance across various scenarios and multiple datasets, proving its generalizability and effectiveness.

## 1 Introduction

In today's information age, digital cameras have become ubiquitous sensing devices across socio-economic domains, serving both human perception and machine interpretation. Based on their objectives, imaging applications divide into two categories: human vision-oriented (e.g., smartphones, social media) and machine vision-oriented Jiao et al. (2023; 2022); Redmon (2018); Xie et al. (2021)(e.g., robotics, medical imaging). Hybrid scenarios requiring dual optimization are emerging but understudied, such as in-vehicle ISPs for autonomous driving that demand real-time visualization for users while enabling environmental recognition for machine navigation. Current research didn't consider dedicated optimization frameworks for such dual-purpose requirements.

In contrast to RGB images, RAW images are directly captured from the sensor before any ISP processing, retaining physically meaningful information such as scene radiance and noise characteristics Wei et al. (2020; 2021). However, due to the substantial volume of RAW data, which entails high storage and transmission costs, along with its suboptimal visualization quality when unprocessed, mainstream approaches in most high-level vision tasks—such as object detection and semantic segmentation—still rely on RGB images as input.

The manually designed Image Signal Processing (ISP) pipeline aims to produce images that offer superior quality for human visual perception Wu et al. (2019). It typically consists of a sequence of operations such as demosaicing, white balance adjustment, color correction, tone mapping, denoising, sharpening, and gamma correction Ramanath et al. (2005). However, each step within the ISP may introduce certain artifacts or degradation in image quality Guo et al. (2024), which can adversely impact the performance of downstream high-level vision tasks. As shown in Fig 1 (a), the conventional processing of RAW data follows a two-stage pipeline: the RAW data is first converted into an RGB image via a human-visual-oriented ISP, and then this RGB image is utilized for tasks such as object detection. It can be observed from the detection results that, due to information loss during ISP processing, the vehicle in the upper-left corner of the image was not detected.

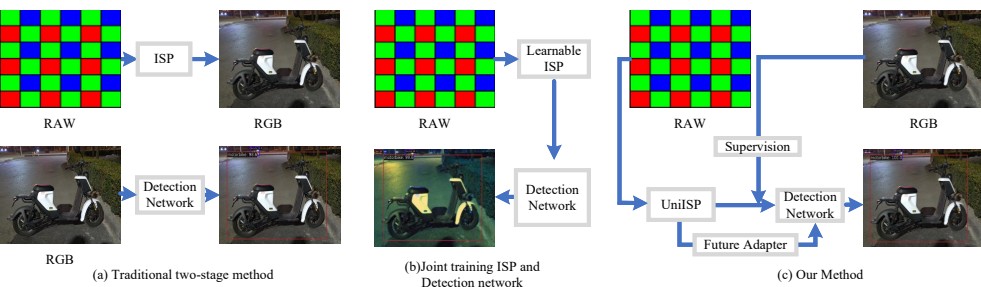

Figure 1: Methods of using RAW data for object detection. (a) Traditional two-stage approaches employ compression operations on the RAW data, resulting in the loss of fine details. (b) Jointly training the ISP and the downstream detection network results in generated RGB images with suboptimal visual quality. (c) Our framework takes into account both the requirements of human visual perception and the needs of machine vision.

To better leverage camera raw data for various computer vision tasks, existing approaches have been proposed, as illustrated in Fig 1 (b). These methods either incorporate differentiable ISP modules Yu et al. (2021) or employ neural networks to replace conventional ISP components Yoshimura et al. (2023). By jointly training the ISP and the downstream network, they directly optimize task-specific performance Wang et al. (2024). Although such strategies improve the performance on downstream tasks, the resulting RGB images produced by the jointly trained ISP often fail to meet human aesthetic standards and may not even be properly visualizable.

To address these limitations, we propose UniISP, a lightweight end-to-end neural ISP framework that simultaneously serves both human and machine vision. Based on a U-Net architecture Ronneberger et al. (2015), it incorporates several carefully designed modules. Inspired by Restormer Zamir et al. (2022), which applies self-attention along feature dimensions to efficiently process high-resolution images, we introduce a Hybrid Attention Module (HAM) based on CNNs. This design preserves the local feature extraction strengths of CNNs while capturing global dependencies via self-attention, maintaining high computational efficiency. Drawing inspiration from feature fusion techniques Lin et al. (2017); Chen et al. (2024), we propose a Feature Adapter to embed rich information from the ISP stage into downstream networks, followed by joint optimization to collectively enhance the performance of final machine vision perception tasks. As illustrated in Fig 1 (c), the HAM, coupled with supervised learning, ensures that the generated images align with human visual preferences, while the Feature Adapter effectively leverages RAW data features for downstream networks, thereby improving detection performance. This dual mechanism enables the framework to simultaneously cater to the requirements of both human visual perception and machine vision. Extensive experiments across multiple tasks, datasets, and scenarios demonstrate that the proposed method outperforms existing methods, confirming its effectiveness and strong generalization capability.

The main contributions of this paper can be summarized as follows: (1) We propose a learnable lightweight ISP framework that can simultaneously meet the needs of human vision and machine vision. (2) We introduce a Hybrid Attention Module, which combines self-attention mechanisms with classic convolutional attention modules, fully leveraging different types of attention mechanisms to enhance the model's feature representation and generalization capabilities. (3) We present a Feature Adapter that integrates and adapts multi-scale features from the ISP stage with downstream networks, enriching the model's understanding ability and improving the performance on downstream tasks. (4) We conduct extensive experiments on a wide range of tasks (including raw-to-RGB mapping, object detection, and semantic segmentation). Qualitative and quantitative results demonstrate that the proposed algorithm achieves state-of-the-art performance.

## 2 RELATED WORK

**Deep Networks for Human-Visual-Quality ISP.** Typically, a camera's ISP pipeline is responsible for reconstructing high-quality sRGB images from raw sensor data. Inspired by the unprecedented success of deep learning, CycleISP Zamir et al. (2020), Invertible-ISP Xing et al. (2021) and ParamISP Kim et al. (2023) proposed a complete forward and inverse camera imaging pipeline. Ignatov et al.

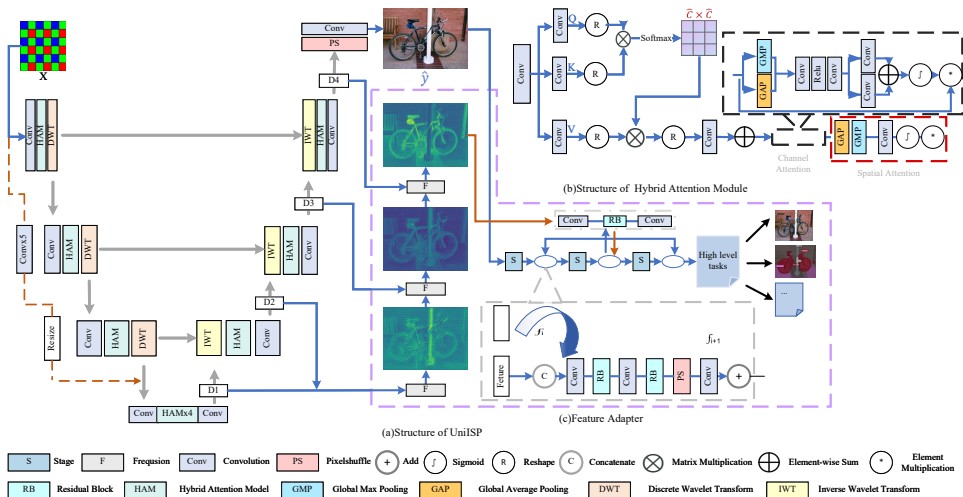

Figure 2: (a) Overall framework of UniISP. Through supervised learning with RGB reference and joint training with downstream networks, it simultaneously optimizes both the perceptual quality of generated RGB images and the performance of downstream tasks. (b) Architecture of the Hybrid Attention Module (HAM). By effectively integrating multiple attention mechanisms, it significantly enhances the model's representational capacity. (c) The feature adapter transfers and adapts RAW-stage features to downstream networks.

(2020b) collected a dataset comprising paired RAW and sRGB images captured by a Huawei P20 smartphone and a Canon 5D Mark IV DSLR camera, respectively. Based on the dataset provided by Ignatov et al. (2020b), two challenges were organized Ignatov et al. (2020a; 2019), where CNN methods inspired by the Multi-Level Wavelet CNN (MWCNN) Liu et al. (2019) achieved the best results. Among the methods based on MWCNN, MW-ISPNet Ignatov et al. (2020a) and AWNet Dai et al. (2020) adopted different U-Net variants to generate visually appealing sRGB images. LiteISPNet Zhang et al. (2021) proposed an alignment loss by explicitly computing the optical flow between the predicted DSLR-like image and the ground truth image. Shekhar Tripathi et al. (2022) introduced a color-conditional ISP network, utilizing a color prediction network with an efficient global context Transformer module to achieve more accurate color prediction.

**Computer Vision based on RAW data.** To more effectively utilize the information in raw images for downstream visual tasks, early methods proposed performing computer vision processing directly on raw images Buckler et al. (2017); Zhou et al. (2020). However, these methods lacked consideration of the physical sensor noise in the conversion process from photons to raw images, especially under low-light conditions Li et al. (2023); Monakhova et al. (2022); Wei et al. (2020). Moreover, training from scratch on raw data would forgo the current visual models pre-trained on large-scale sRGB data, especially since existing raw image datasets Omid-Zohoor et al. (2014); Zhou et al. (2017) are far fewer than RGB datasets Deng et al. (2009); Kirillov et al. (2023). Therefore, subsequent research has mainly focused on finding methods to jointly optimize the ISP and backend computer vision models Diamond et al. (2021); Mosleh et al. (2020); Qin et al. (2022). Qin et al. (2023) designed a sequential CNN model that repeatedly adjusts the hyperparameters of the ISP to adapt to downstream tasks, demonstrating the advantages of this approach. Cui et al. (2021) considered the physical sensor noise model in the ISP and proposed a multi-task auto-encoder transformation model to learn intrinsic visual structures, proving its effectiveness in object detection under dark environments. Guo et al. (2024) improved the generalization ability and performance on downstream tasks by learning degradation-independent latent representations for the ISP. Furthermore, RAW-Adapter Cui & Harada (2025) enhances the performance on downstream tasks by introducing learnable ISP stages and model-level adapters to adapt pre-trained sRGB models to the camera's RAW data.

## 3 METHOD

Fig 2 illustrates the overall architecture of the UniISP model and its constituent modules. This paper will first detail the key modules designed to enhance human visual quality (section 3.1). Subsequently,

we introduce the components incorporated to improve perceptual performance for downstream tasks (section 3.2). Finally, we elaborate on the adaptive training framework designed for the joint optimization of these multi-task objectives.

## 3.1 HUMAN-VISION-ORIENTED ISP MODULE

In the context of human visual perception tasks, this paper endeavors to implement a lightweight, deep learning-based ISP module. Given a raw image $x$ captured by a smartphone camera, the goal is to predict a high-quality sRGB image $\hat{y}$ that approximates the color characteristics of a target DSLR sRGB image $y$. we propose UniISP, a multi-level wavelet-based Liu et al. (2019) ISP network based on the U-Net architecture with the HAM as its backbone. The overall architecture of the UniISP model is shown in Fig 2.

**Hybrid Attention Module (HAM):** Our HAM is designed for efficient and feature-rich global modeling, which is essential for high-quality image reconstruction. Inspired by Restormer Zamir et al. (2022), we implement self-attention (SA) along the channel dimension rather than the conventional spatial dimension. This reduces computational complexity from $\mathcal{O}\left((H \cdot W)^2\right)$ to $\mathcal{O}(C^2)$ for feature maps of size $H \times W \times C$, making global interactions feasible for high-resolution images.

Channel-wise SA enables the model to capture and reinforce semantic correlations between color channels, which are crucial for RAW-to-RGB conversion and related tasks. Unlike static channel attention methods such as Squeeze-and-Excitation Hu et al. (2018), our approach dynamically amplifies informative channels and suppresses noisy ones according to context.

To preserve structural information, we further integrate relative positional encoding (RPE) into the channel-wise SA. RPE provides cues about the relative positions of channels, enhancing the modeling of both local and long-range dependencies. As shown in Fig 2(b), the input tensor is projected into a higher-dimensional channel space, partitioned into query, key, and value tensors, and processed with channel-wise SA augmented by RPE. The resulting feature map is globally informed by both channel content and relationships, and is further refined by subsequent attention mechanisms for improved task performance.

Since $x$ and $y$ are captured by different cameras, there is inevitably a spatial misalignment. Furthermore, the severe color discrepancies between $x$ and $y$ make the image alignment more challenging. Therefore, this paper incorporates the global color mapping(GCM) module, an alignment method proposed by LiteISP Zhang et al. (2021), for joint training and optimization. To address inaccuracies introduced during the alignment process, a robust mask alignment loss is proposed by enhancing optical flow consistency checks.

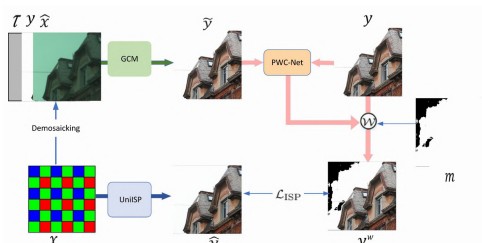

Figure 3: Joint training with GCM. The well-aligned supervisory target sRGB image $y^w$ is synthesized through Global Color Mapping (GCM) and optical flow consistency mask $m$ to enforce spatiotemporal alignment constraints during training.

As shown in Fig 3, the input $x$ first undergoes a simple demosaicing method (*e.g.*, bicubic) to produce $\hat{x}$. $\tau \in \mathbb{R}^{2 \times H \times W}$ is the 2D coordinate map containing the coordinate of the pixels,which is normalized to $[-1, 1]$. The $\tau$,$\hat{x}$ and $y$ pass through the GCM module to obtain a color-adjusted $\tilde{y}$. Given $\tilde{y}$ and $y$, a pre-trained optical flow network, PWC-NETSun et al. (2018), is employed to evaluate the optical flow between the two images. The estimated optical flow is then used to perform a warping operation $\mathcal{W}$ Sun et al. (2018) on $y$ to form a well-aligned target sRGB image $y^w$.

The use of optical flow estimation significantly alleviates misalignment issues; however, the method itself has some limitations. In particular, optical flow is often inaccurate in the presence of repeating patterns, occlusions, and homogeneous regions. To address these challenges, we propose a more robust and flexible alignment mask that employs forward-backward consistency checks to filter out inaccurate areas.

For the pixel position $(x, y)$ in frame $I_A$, $(u, v)$ represents the displacement from $(x, y)$ in frame $I_A$ to the corresponding position in frame $I_B$, where $\Psi$ is the pre-trained optical flow network .

$$
\begin{aligned}
(u, v) &= \Psi(a, b)(x, y) \\
(x', y') &= (x + u, y + v) \\
(u', v') &= \Psi(b, a)(x', y')
\end{aligned}
\tag{1}
$$

$\boldsymbol{m}$ represents the optical flow consistency mask. Here, each element $m_{x,y}$ of $\boldsymbol{m}$ is defined as follows:

$$
m_{x,y} = \begin{cases} 1, & \sqrt{(u + u')^2 + (v + v')^2} < T, \\ & \text{and} \quad \mathcal{W}(1, (u, v)) \geq 1 - \epsilon \\ 0, & \text{otherwise} \end{cases}
\tag{2}
$$

Here, $T$ is set to the 90th percentile and can be adjusted flexibly. $\epsilon$ is a threshold set to 0.001.

The loss function for human visual tasks on unaligned datasets is as follows:

$$
\begin{aligned}
L_{\text{ISP}}(\hat{\mathbf{y}}, \mathbf{y}^w) = {} & \lambda_1 \left\| \mathbf{m} \circ (\hat{\mathbf{y}} - \mathbf{y}^w) \right\|_1 + \lambda_2 (1 - L_{\text{SSIM}}(\hat{\mathbf{y}}, \mathbf{y}^w)) \\
& + \lambda_3 \left\| \mathbf{m} \circ (\phi(\hat{\mathbf{y}}) - \phi(\mathbf{y}^w)) \right\|_1
\end{aligned}
\tag{3}
$$

Here, $\circ$ denotes the element-wise product, and $\| \cdot \|_1$ is the L1 loss. $L_{\text{SSIM}}$ represents the structural similarity (SSIM) loss Wang et al. (2003). The function $\phi$ denotes the pre-trained VGG-19 network Mao et al. (2017). We set the parameters $\lambda_1 = \lambda_3 = 1$ and $\lambda_2 = 0.15$.

## 3.2 FEATURE ADAPTER

To fully exploit the rich information contained in RAW images and enable effective adaptation to downstream RGB-pretrained models, our design incorporates a feature adapter based on frequency-domain fusion and multi-stage integration.

RAW data preserves both fine-grained details (high-frequency information) and global context (low-frequency information) that are often lost in traditional ISP pipelines. Directly mapping RAW features to RGB without considering these frequency characteristics can lead to suboptimal representation and poor transferability to downstream tasks. By fusing multi-scale encoder outputs (D1–D4), FreqFusion module Chen et al. (2024) aggregates information across spatial resolutions and frequency bands, ensuring that features relevant for both reconstruction and semantic analysis are retained.

See Fig 2, the FreqFusion module aggregates multi-scale encoder outputs and uses spectral processing to separate low and high-frequency components. The fused feature is computed as:

$$
F_{\text{fused}} = \sum_{i=1}^{3} \left( \alpha_i F_{\text{low}}^i + \beta_i F_{\text{high}}^i \right)
\tag{4}
$$

where $\alpha_i$ and $\beta_i$ are learnable weights. This approach preserves both boundary details and global structure. The fused features are transformed and progressively merged into the backbone of the downstream network at multiple stages. This merging is realized via concatenation and residual connections:

$$
f_{l+1} = \text{ResBlock}\left(\text{Concat}(f_l, F_{\text{fused}})\right) + f_l
\tag{5}
$$

where $f_l$ is the backbone feature at stage $l$. This mechanism enables the network to flexibly incorporate both RAW-domain and RGB-domain information, reducing domain shift and facilitating effective transfer of pre-trained weights.

To balance human visual quality and machine perception performance, we propose an adaptive total loss $\mathcal{L}_{\text{total}}$ that dynamically combines human vision loss $\mathcal{L}_{\text{human}}$ and machine vision loss $\mathcal{L}_{\text{machine}}$:

$$
\mathcal{L}_{\text{total}} = \lambda \cdot \mathcal{L}_{\text{human}} + (1 - \lambda) \cdot \mathcal{L}_{\text{machine}}
\tag{6}
$$

The weighting factor $\lambda \in [0, 1]$ is adaptively computed at each step $t$ based on exponential moving averages of the losses:

$$
\lambda^{(t)} = \frac{\widetilde{\mathcal{L}}_{\text{machine}}^{(t)}}{\widetilde{\mathcal{L}}_{\text{human}}^{(t)} + \widetilde{\mathcal{L}}_{\text{machine}}^{(t)}}
\tag{7}
$$

where the exponentially moving average (EMA) losses are updated as:

$$\widetilde{\mathcal{L}}_{\text{human}}^{(t)} = \gamma \cdot \widetilde{\mathcal{L}}_{\text{human}}^{(t-1)} + (1-\gamma) \cdot \mathcal{L}_{\text{human}}^{(t)} \tag{8}$$

$$\widetilde{\mathcal{L}}_{\text{machine}}^{(t)} = \gamma \cdot \widetilde{\mathcal{L}}_{\text{machine}}^{(t-1)} + (1-\gamma) \cdot \mathcal{L}_{\text{machine}}^{(t)} \tag{9}$$

with $\gamma = 0.9$. This design automatically adjusts focus: when $\widetilde{\mathcal{L}}_{\text{human}}$ is relatively high, $\lambda$ decreases to emphasize machine performance, and vice versa.

$\mathcal{L}_{\text{machine}}$ is defined by the downstream task objective, e.g., cross-entropy loss for segmentation or focal loss for detection. For human vision, $\mathcal{L}_{\text{human}}$ is defined as $\mathcal{L}_{\text{ISP}}(\hat{\mathbf{y}}, \mathbf{y}^w)$ if the training dataset is unaligned, or a combination of $\mathcal{L}_1$ and $\mathcal{L}_{\text{SSIM}}$ losses if aligned.

## 4 EXPERIMENTS

### 4.1 DATASET AND IMPLEMENTATION DETAILS

**Datasets.** We conducted experiments on raw-to-RGB mapping, object detection, Dark light enhancement, and semantic segmentation tasks. An overview of the datasets is shown in Table 1.

Table 1: Datasets used in the comparison experiments. Here, we synthesize dark and over-exposed images from the original real-captured PASCAL RAW to enrich data diversity. ADE20K RAW is synthesized from ADE20K sRGB via InvISP with dark/over-exposed variants.

| Dataset | # Images | Task | Type | Sensor |
|---|---|---|---|---|
| ZRR | 48,043 | Raw-to-RGB Mapping | Real | Huawei P20, Canon 5D Mark IV |
| PASCAL RAW | 4259 | Object Detection | Real & Synthesis | Nikon D3200 |
| LOD | 2230 | Object Detection | Real | Cannon EOS 5D Mark IV |
| ADE20K RAW | 27,574 | Semantic Segmentation | Synthesis | — |
| SID | 5094 | Dark light enhancement | Real | Sony$\alpha$7SII, Fujifilm X-T2 |
| NOD | 7200 | Object Detection | Real | Sony RX100 VII, Nikon D750 |

In the raw-to-RGB mapping task, we used the ZRR dataset Ignatov et al. (2020b). The ZRR dataset contains 46.8k RAWsRGB image pairs for training and 1.2k pairs for evaluation, where the RAW images are captured by Huawei P20 and the sRGB images are captured by Canon camera.

In the dark light enhancement task, we used the SID dataset Chen et al. (2018). The See-in-the-Dark (SID) dataset contains 5,094 raw short-exposure images, each with a corresponding long-exposure reference image. The images were captured using two cameras: Sony $\alpha$7S II and Fujifilm X-T2.

For the object detection task, we used three open-source real-world datasets: PASCAL RAW Omid-Zohoor et al. (2014), LOD Hong et al. (2021) and NOD Morawski et al. (2022).

PASCAL RAW is a dataset collected during daytime scenes in Palo Alto and San Francisco, containing 4,259 raw images captured by a Nikon D3200 DSLR camera, featuring three object detection classes. For PASCAL RAW (dark) and PASCAL RAW (over-exp), the scene brightness data was generated using the synthesis method provided by Cui & Harada (2025).

LOD is a real-world dataset with 2230 low-light condition RAW images, taken by a Canon EOS 5D Mark IV camera with 8 object classes. We take 1800 images as training set and the other 430 images as test set.

The NOD dataset includes 14-bit RAW outdoor images taken under low-light conditions. It includes 7,200 images, with 3,200 captured by the Sony RX100 VII (NOD-Sony) and 4,000 by the Nikon D750 (NOD-Nikon).The dataset is annotated with bounding boxes for 46,000 instances of people, bicycle, and car classes.

For the semantic segmentation task, we employed ADE20K RAW Cui & Harada (2025), which was synthetically derived from the ADE20K dataset Zhou et al. (2017). The training and testing splits are the same as those of ADE20K.

**Implementation Details.** For the human visual raw-to-RGB mapping task, all ISP models use the GCM module to align GT with raw before proceeding with training and testing. We employed the Adam optimizer Kingma (2014) for optimization, with $\beta_1 = 0.9$ and $\beta_2 = 0.999$, and trained for

100 epochs. The batch size was set to 16, the initial learning rate was set to $1 \times 10^{-4}$, and we used cosine annealing as the learning rate adjustment strategy.

For downstream computer vision tasks, we built our framework based on the open-source computer vision toolkits MMDetection Chen et al. (2019) and MMSegmentation Contributors (2020). Both the object detection and semantic segmentation tasks were initialized with ImageNet pre-trained weights. For object detection tasks, we employed two mainstream detectors: RetinaNet Lin (2017) and Sparse-RCNN Sun et al. (2021), both using ResNet He et al. (2016) as the backbone. For semantic segmentation tasks, we opted for the mainstream segmentation framework Segformer Xie et al. (2021) with MIT Xie et al. (2021) as the backbone. All training image data underwent data augmentation, primarily consisting of random cropping and random flipping, with all experiments conducted on 8 NVIDIA A100 GPUs.

## 4.2 RAW-TO-RGB MAPPING

To validate the effectiveness of the proposed method, comparisons were made against three state-of-the-art methods AWNet Dai et al. (2020), MWISPNet Ignatov et al. (2019), LiteISP Zhang et al. (2021), and FourierISP He et al. (2024) on the ZRR dataset. For quantitative performance evaluation, three metrics were computed on the RGB channels: Peak Signal-to-Noise Ratio (PSNR), Structural Similarity Index (SSIM), and Learned Perceptual Image Patch Similarity (LPIPS) Zhang et al. (2018). Additionally, the number of model parameters and computational complexity were measured to evaluate the efficiency of the models.

As shown in Table 2, UniISP outperforms competing methods across all metrics. Since UniISP primarily comprises the lightweight HAM module, it maintains significantly lower Params and GFLOPs compared to other competing methods. For better visualization, we provide the error maps beneath each corresponding image that indicate the content discrepancies between the generated sRGB images and GT images. Moreover, we present qualitative results

Table 2: Raw-to-RGB mapping results on the ZRR (Align GT with RAW) dataset. UniISP (w/o F) denotes the variant without the feature adapter module.

| Method | Params (M) | GFLOPs | PSNR↑ | SSIM↑ | LPIPS↓ |
|---|---|---|---|---|---|
| PyNet | 47.53 | 342.42 | 22.93 | 0.8509 | 0.153 |
| MW-ISPNet | 29.22 | 88.39 | 23.28 | 0.8524 | 0.148 |
| AWNet | 49.07 | 75.08 | 23.46 | 0.8537 | 0.141 |
| LiteISP | 9.01 | 71.94 | 23.88 | 0.8573 | 0.134 |
| FourierISP | 6.18 | 106.40 | 23.98 | 0.8592 | 0.128 |
| UniISP (w/o F) | **5.04** | **31.68** | **24.14** | **0.8614** | **0.122** |

in Fig 7. From the figure, it is evident that the proposed method yields sharper images with richer details and more accurate colors, compared to competing methods.

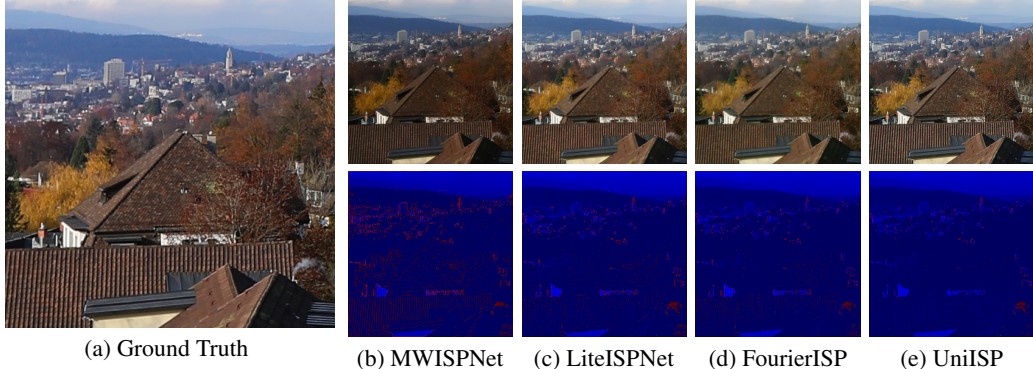

(a) Ground Truth    (b) MWISPNet    (c) LiteISPNet    (d) FourierISP    (e) UniISP

Figure 4: Visual results comparison of a typical scene in the ZRR dataset. The error maps represent the content difference between the generated sRGB images and the GT images, the darker the better. Best viewed by zooming in.

## 4.3 OBJECT DETECTION

For the object detection task on the PASCAL RAW Omid-Zohoor et al. (2014) dataset, we adopt RetinaNet Lin (2017) with ResNet He et al. (2016) backbones of varying sizes (ResNet-18, ResNet-

Table 3: Object detection results (VOC-style mAP) on the PASCAL RAW and LOD datasets using ResNet-18 (R-18) and ResNet-50 (R-50) backbones. Best scores are highlighted in **bold.**

| Dataset | PASCAL RAW | | | | | | LOD | |
|---|---|---|---|---|---|---|---|---|
| | Normal | Over-exposed | Dark | Normal | Over-exposed | Dark | | |
| Detecter | RetinaNet (R-18) | | | RetinaNet (R-50) | | | RetinaNet (R-50) | Sp-RCNN (R-50) |
| Default ISP | 88.3 | - | - | 89.6 | - | - | 58.4 | 53.9 |
| Demosaicing | 87.7 | 87.7 | 80.3 | 89.2 | 88.8 | 82.6 | 58.5 | 57.7 |
| Karaimer *et al.* | 88.1 | 85.6 | 78.8 | 89.4 | 86.8 | 79.6 | 54.4 | 52.2 |
| LiteISP | 85.2 | 84.2 | 71.9 | 88.5 | 85.1 | 73.5 | 55.3 | 49.3 |
| InvISP | 85.4 | 86.6 | 70.9 | 87.6 | 87.3 | 74.7 | 56.9 | 49.4 |
| SID | - | - | 78.2 | - | - | 81.5 | 49.1 | 43.1 |
| DNF | - | - | 81.1 | - | - | 82.8 | - | - |
| Dirty-Pixel | 88.6 | 88.0 | 80.8 | 89.7 | 89.0 | 83.6 | 61.6 | 58.8 |
| RAW-Adapter | 88.7 | 88.7 | 82.5 | 89.7 | 89.5 | 86.6 | 62.1 | 59.2 |
| UniISP(w/o J) | 88.7 | 86.2 | 79.2 | 89.7 | 87.6 | 82.5 | 55.8 | 50.8 |
| UniISP(w/o F) | 89.0 | 88.4 | 83.7 | 89.8 | 89.5 | 86.8 | 62.4 | 60.2 |
| UniISP | **89.1** | **89.0** | **85.2** | **89.8** | **89.7** | **88.0** | **63.9** | **62.3** |

50). All models were trained using the SGD optimizer Ruder (2016) with a batch size of 4. The training images were cropped to a size of (400, 600), and the training was conducted for 50 epochs. Table 3 presents the detection results with ResNet-18 and ResNet-50 backbones, The comparison includes demosaiced RAW data ("Demosaicing"), the default camera ISP, various state-of-the-art ISP solutions Chen et al. (2018); Jin et al. (2023); Karaimer & Brown (2016); Zhang et al. (2021); Xing et al. (2021), and jointly optimized methods including Dirty-Pixel Diamond et al. (2021) and RAW-Adapter Cui & Harada (2025). Among these, UniISP (w/o J) denotes the model without the feature adapter and without joint optimization with downstream tasks, using the weights trained on the ZRR dataset. UniISP (w/o F) represents the model jointly trained and optimized with the downstream network but without the feature adapter.

The advantage of raw data compared to RGB data lies in its ability to provide a richer representation under challenging conditions such as low-light environments. From the Table 3, it is evident that the proposed method significantly improves performance in dark scenes compared to the RAW-Adapter Cui & Harada (2025). This indicates that by using the feature adapter module, our approach better leverages the ISP-stage feature adaptation for downstream networks. By evaluating the performance of UniISP (w/o F) in low-light scenarios through ablation experiments, we further demonstrate the effectiveness of the feature model.

While the quantitative improvements in over-exposed conditions are relatively modest (0.2% mAP gain), UniISP produces images with superior human visual quality, avoiding the harsh contrast and blown-out highlights common in machine-only optimization approaches, thus achieving a better balance between human perception and machine performance. Overall, the proposed method surpasses other competing methods under various lighting conditions. To further validate the performance of the proposed method in low-light environments, experiments were conducted on the LOD dataset. We employed two detectors, RetinaNet and Sparse-RCNN, both utilizing a ResNet-50 backbone. As shown in the Table 3, the method proposed in this paper significantly enhances performance in low-light scenarios.

As illustrated in Fig 5, the visualization results of object detection under various illumination conditions in the PASCAL RAW dataset are presented. Notably, UniISP (w/o J) generates images with superior visual quality compared to other methods, further demonstrating its strong generalization capability across diverse raw inputs in the raw-to-RGB conversion task. The figure indicates that UniISP achieves robust detection performance across varying lighting environments, while other methods exhibit varying degrees of missed or false detections.

Furthermore, since UniISP incorporates a human vision-oriented loss during training and utilizes images generated by UniISP (w/o J) as ground truth, the resulting images not only align with human visual perception but also bridge the domain gap between raw sensor data and the RGB domain preferred by pre-trained backbones. Moreover, the feature adapter effectively propagates feature information from the raw domain to downstream networks, thereby further enhancing perceptual performance. The experimental results validate the feasibility of joint optimization for both human vision and machine vision through a multi-task learning framework. More visualizations and cross-sensor generalization experiments can be found in Appendix C.

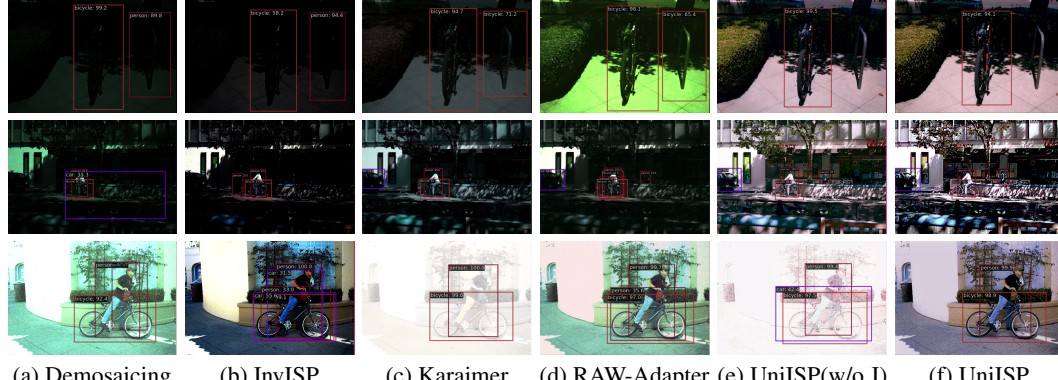

| (a) Demosaicing | (b) InvISP | (c) Karaimer | (d) RAW-Adapter | (e) UniISP(w/o J) | (f) UniISP |

Figure 5: Visualization of object detection results on PASCAL RAW. Three rows represent dark, normal and over-exposed scenarios respectively (from top to bottom). Our UniISP obtains higher detection accuracy while maintaining good visual quality.

## 4.4 SEMANTIC SEGMENTATION

For semantic segmentation on ADE20K RAW dataset, we choose Segformer as the segmentation framework with the different size MIT backbones (MITB5, MIT-B3, MIT-B0), all the models are trained on 4 NVIDIA Tesla A100 GPUs with Adam optimizer, the batch size is set to 4, training images are cropped into $512 \times 512$ and training iteration number is set to 80000. Comparison results are shown in Table 4, where we compare both efficiency (parameters, inference time) and performance (mIOU). Inference time is calculated on a single Tesla A100 GPU. From the table, it can be observed that the proposed method in this paper outperforms other competing methods across multiple illumination scenarios with different backbone sizes. In low-light conditions, thanks to the feature adapter module, UniISP with a less parameterized backbone (MIT-B3) even surpasses other methods that use a more-parameter backbone (MIT-B5). Since the feature adapter fuses features from the ISP stages through multiple FreqFusion operations, it does not have an advantage in inference time.

Table 4: Semantic segmentation results on the ADE20K RAW dataset. Results of three brightness levels are listed in separate columns. We also show parameter number and inference time of each method. Best scores are highlighted in **bold**.

| Method | Backbone | Params(M) ↓ | Inference time(s) ↓ | mIOU ↑ | | |
|---|---|---|---|---|---|---|
| | | | | Normal | Over-exposed | Dark |
| Demosaicing | | | 0.085 | 47.47 | 45.69 | 37.55 |
| Karaimer *et al.* | | | 0.291 | 45.48 | 42.85 | 37.32 |
| InvISP | | | 0.145 | 47.82 | 44.30 | 4.03 |
| LiteISP | MIT-B5 | 82.01 | 0.187 | 43.22 | 42.01 | 5.52 |
| DNF | | | 0.138 | - | - | 35.88 |
| SID | | | 0.208 | - | - | 37.06 |
| UniISP(w/o J) | | | 0.152 | 46.38 | 44.37 | 37.42 |
| | MIT-B5 | 86.29 | 0.119 | 47.86 | 46.50 | 38.02 |
| Dirty-Pixel | MIT-B3 | 48.92 | 0.048 | 46.19 | 44.13 | 36.93 |
| | MIT-B0 | 8.00 | **0.031** | 34.43 | 31.10 | 24.89 |
| | MIT-B5 | 82.31 | 0.121 | 47.95 | 46.62 | 38.75 |
| RAW-Adapter | MIT-B3 | 45.16 | 0.052 | 46.57 | 44.19 | 37.62 |
| | MIT-B0 | **3.87** | 0.038 | 34.72 | 31.91 | 25.06 |
| | MIT-B5 | 88.04 | 0.185 | 47.94 | 46.60 | 38.13 |
| UniISP(w/o F) | MIT-B3 | 50.49 | 0.161 | 46.65 | 44.26 | 37.53 |
| | MIT-B0 | 9.37 | 0.151 | 34.83 | 32.04 | 25.23 |
| | MIT-B5 | 89.47 | 0.238 | **48.04** | **46.65** | **38.84** |
| UniISP | MIT-B3 | 51.92 | 0.217 | 46.70 | 44.71 | 38.37 |
| | MIT-B0 | 10.84 | 0.192 | 35.02 | 32.18 | 26.08 |

Fig 6 visualizes the semantic segmentation results, where UniISP demonstrates superior boundary precision and contour clarity compared to baseline methods.This improvement is attributed to the feature adapter, which explicitly decomposes raw sensor data into multi-scale frequency components via learnable bandpass filters, followed by cross-frequency feature fusion that selectively enhances high-frequency structural cues (e.g., edges, textures) while suppressing noise.

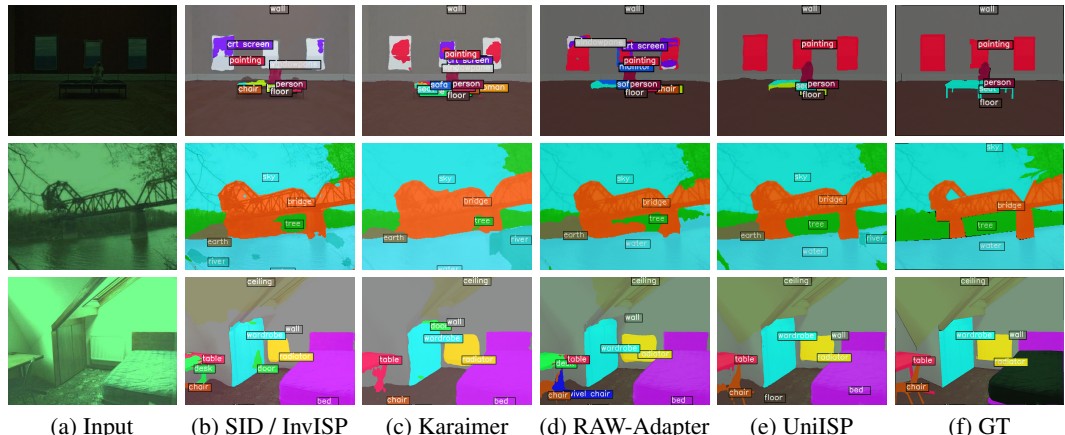

|(a) Input|(b) SID / InvISP|(c) Karaimer|(d) RAW-Adapter|(e) UniISP|(f) GT|

Figure 6: Visualization of semantic segmentation results on ADE20K RAW. Three rows represent dark, normal and over-exposed scenarios respectively (from top to bottom).

## 5 CONCLUSION

This work presents the first exploration of simultaneously optimizing Image Signal Processing (ISP) for both human visual perception and machine vision tasks, proposing a novel framework named UniISP. Firstly, we introduce a lightweight module called HAM (Hybrid Attention Module), which integrates multiple attention mechanisms to enhance feature representation and improve model generalization during the ISP stage. Secondly, to better adapt the features of raw data to pre-trained sRGB models in downstream computer vision tasks, we propose a feature adapter. Extensive experiments on multiple datasets across different tasks demonstrate that UniISP achieves state-of-the-art results. This work suggests potential directions for optimizing future ISP systems.

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

## A    RAW-TO-RGB MAPPING

A qualitative comparison of the proposed method and existing approaches on the ZRR test set is provided in Fig 7. As depicted, earlier methods including MWISPNet and AWNet exhibit difficulties in preserving color accuracy and fine-grained details. While LiteISPNet and FourierISP demonstrates some improvement, it remains constrained in rendering rich textures and ensuring consistent color reproduction, especially in areas with complicated illumination or fine structures. These shortcomings frequently result in noticeable artifacts or the omission of critical visual content.

By comparison, our approach yields reconstructions that closely align with the ground truth in terms of visual quality. The method faithfully reconstructs delicate patterns and subtle gradients, while achieving vibrant and color-accurate results. Exhibiting visibly sharper edges, more refined textures, and more natural color rendition, our outputs outperform those of other methods. Notably, UniISP substantially mitigates typical artifacts—such as excessive smoothing, color inaccuracies, and loss of detail—that commonly affect existing techniques.

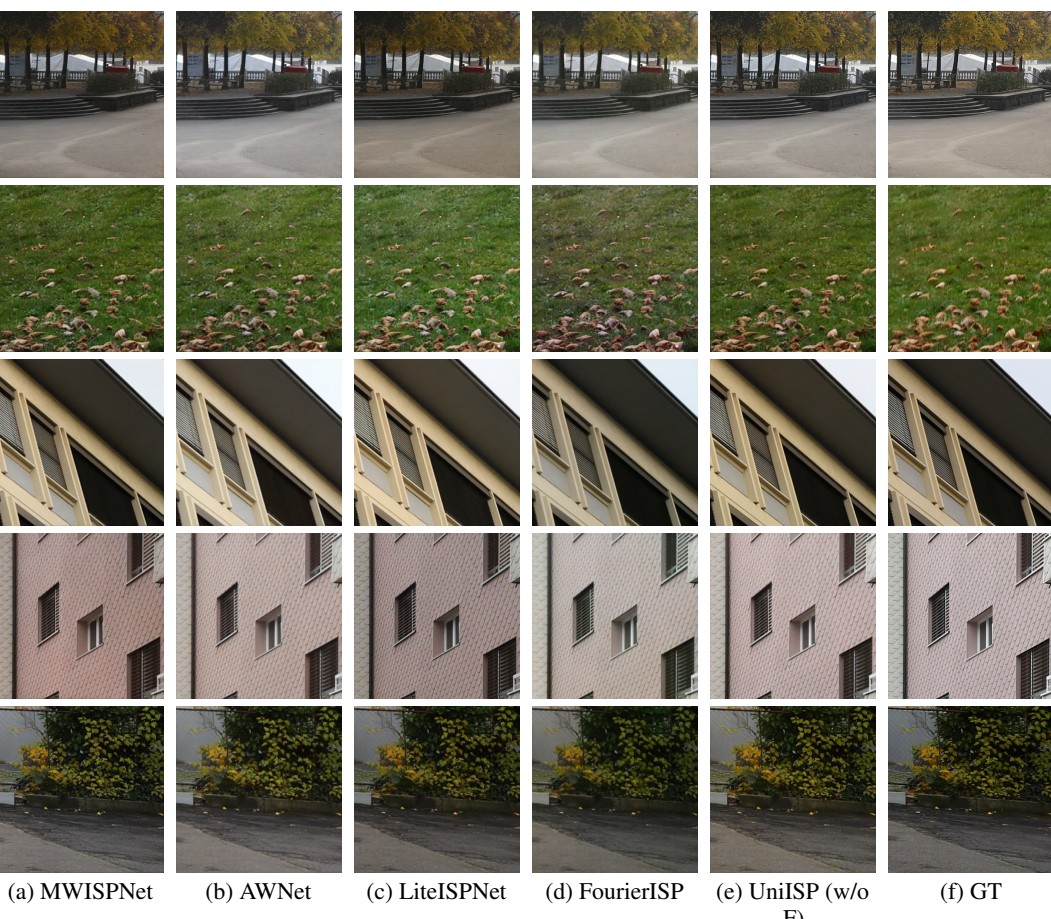

(a) MWISPNet  (b) AWNet  (c) LiteISPNet  (d) FourierISP  (e) UniISP (w/o F)  (f) GT

Figure 7: Visual comparison of RAW-to-RGB results on the ZRR dataset.

## B    DARK LIGHT ENHANCEMENT

To comprehensively evaluate the generalization capability of our proposed method, we conducted extensive experiments on the challenging SID-Sony dataset Chen et al. (2018), which contains real-world extremely low-light RAW images. We followed the official train-test split provided by the SID dataset, where each pair includes a short-exposure input RAW and its long-exposure reference. The network was trained with L1 loss, Adam optimizer (learning rate: $1 \times 10^{-4}$, batch size: 8) for 500 epochs, with learning rate decay at epoch 400.

Table 5: Comparison of different methods on SID-Sony datasets

| Method | Params(M) | GFLOPs | x100 | | | x300 | | |
|---|---|---|---|---|---|---|---|---|
| | | | PSNR ↑ | SSIM ↑ | LPIPS ↓ | PSNR ↑ | SSIM ↑ | LPIPS ↓ |
| SID | 7.76 | **10.50** | 27.86 | 0.7538 | 0.6287 | 26.53 | 0.6802 | 0.7088 |
| DNF | **2.83** | 12.55 | 28.20 | 0.7671 | 0.5044 | 26.81 | 0.7157 | 0.5828 |
| LiteISP | 9.01 | 71.94 | 29.68 | 0.785 | 0.470 | 28.01 | 0.720 | 0.5430 |
| UniISP (w/o F) | 5.04 | 31.68 | **30.31** | **0.7982** | **0.4228** | **28.49** | **0.7393** | **0.5108** |

Table 5 presents a comprehensive compares our method with other approaches under different exposure ratios (×100 and ×300), where the ratio indicates the exposure time of reference images relative to input images. UniISP delivers superior performance, achieving substantial improvements over efficient methods like SID and DNF. Additionally, compared to LiteISP, it demonstrates higher execution efficiency. The consistent performance across different exposure ratios validates the robustness and generalization capability of our unified ISP architecture in extreme low-light conditions.

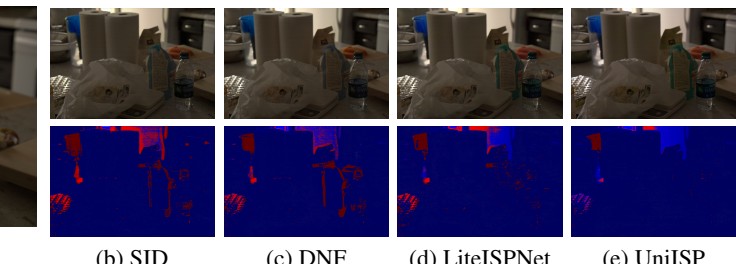

(a) Ground Truth      (b) SID      (c) DNF      (d) LiteISPNet      (e) UniISP

Figure 8: Visual comparison on the SID-Sony dataset. The error maps represent the content difference between the generated sRGB images and the GT images, the darker the better. Best viewed by zooming in.

Fig 8 shows visual results on SID-Sony dataset (×100 ratio) with corresponding error maps below. Error maps clearly demonstrate UniISP superiority—achieving minimal reconstruction error (darkest map) compared to baselines significant noise and color distortion (bright regions), validating our method's effectiveness in extreme low-light imaging.

## C OBJECT DETECTION

### C.1 GENERALIZATION ACROSS DIFFERENT SENSORS

To evaluate the generalization capability of our method across different camera sensors, we conducted experiments on the real-world NOD dataset Morawski et al. (2022), which includes data from both Sony and Nikon sensors. Table 6 presents the object detection performance (mAP) under different configurations. UniISP(Sony), trained only on Sony data, achieves 32.1 mAP on Nikon images without fine-tuning—only 2.1 points below the RAW baseline, demonstrating reasonable cross-sensor transferability. When trained on target Nikon data, UniISP(Nikon) achieves the best performance.

Table 6: Cross-sensor generalization on NOD dataset. UniISP(Sony): trained on Sony, tested on Nikon; Others: trained and tested on Nikon.

| Method | RAW | RGB | RAW-Adapter | UniISP(Sony) | UniISP(Nikon) |
|---|---|---|---|---|---|
| mAP | 34.2 | 34.5 | 36.3 | 32.1 | 38.4 |

Figure 9 visualizes the cross-sensor generalization performance. The RGB baseline shows missed detections, while RAW-Adapter exhibits false detections. UniISP(Sony), trained only on Sony data, successfully processes Nikon RAW images with acceptable quality, detecting most objects despite the sensor domain gap. UniISP(Nikon) trained on target data achieves the best results with natural colors and complete detection coverage. This demonstrates our method's robust cross-sensor transferability.

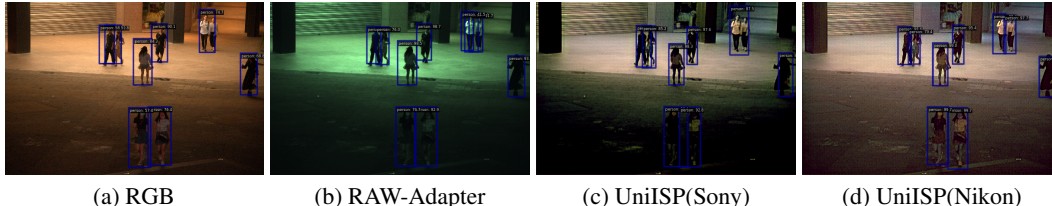

| (a) RGB | (b) RAW-Adapter | (c) UniISP(Sony) | (d) UniISP(Nikon) |

Figure 9: Cross-sensor generalization on NOD-Nikon dataset. UniISP(Sony) trained on Sony data shows reasonable performance on Nikon images, while UniISP(Nikon) achieves optimal results.

## C.2 ADDITIONAL VISUALIZATION

As Fig 10 demonstrates, the proposed method surpasses existing approaches in both detection performance and subjective visual quality, while confirming the feasibility of simultaneously optimizing metrics for both human visual perception and machine vision.

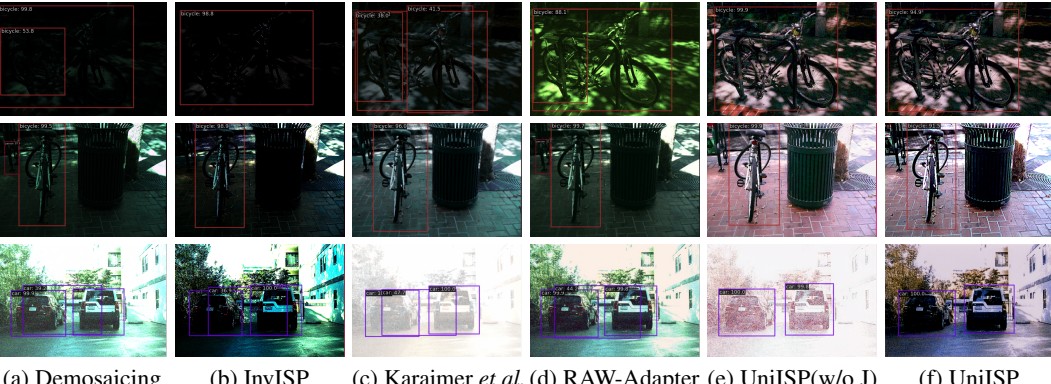

| (a) Demosaicing | (b) InvISP | (c) Karaimer *et al.* | (d) RAW-Adapter | (e) UniISP(w/o J) | (f) UniISP |

Figure 10: Visualization of object detection results on PASCAL RAW. Three rows represent dark, normal and over-exposed scenarios respectively (from top to bottom).

The proposed UniISP framework uniquely optimizes RAW data processing under extremely low-light conditions for both human aesthetics and machine perception. Supervised loss guides the HAM module to ensure high-quality RAW-to-RGB outputs that align with human visual perception, while simultaneously ensuring alignment between the raw domain images and the RGB domain preferred by pre-trained backbones. Meanwhile, the Feature Adapter module leverages raw domain features to facilitate downstream detection tasks. UniISP coordinates both the HAM and FA modules to serve these dual objectives, achieving superior visual quality and robust object detection performance. The effectiveness of the method is further validated on the challenging real-world LOD dataset, as shown in Fig 11. Its generalization capability is also demonstrated across both the PASCAL RAW and LOD datasets.

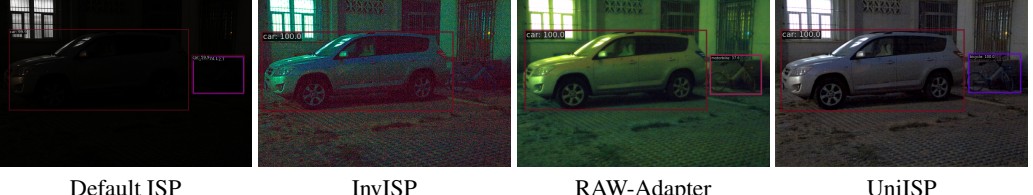

| Default ISP | InvISP | RAW-Adapter | UniISP |

Figure 11: Experimental and visual comparisons under the real extremely dark dataset LOD.

## D SEMANTIC SEGMENTATION

As shown in Fig 12, our proposed UniISP consistently delivers superior semantic segmentation results across challenging illumination conditions, including dark, normal, and over-exposed RAW

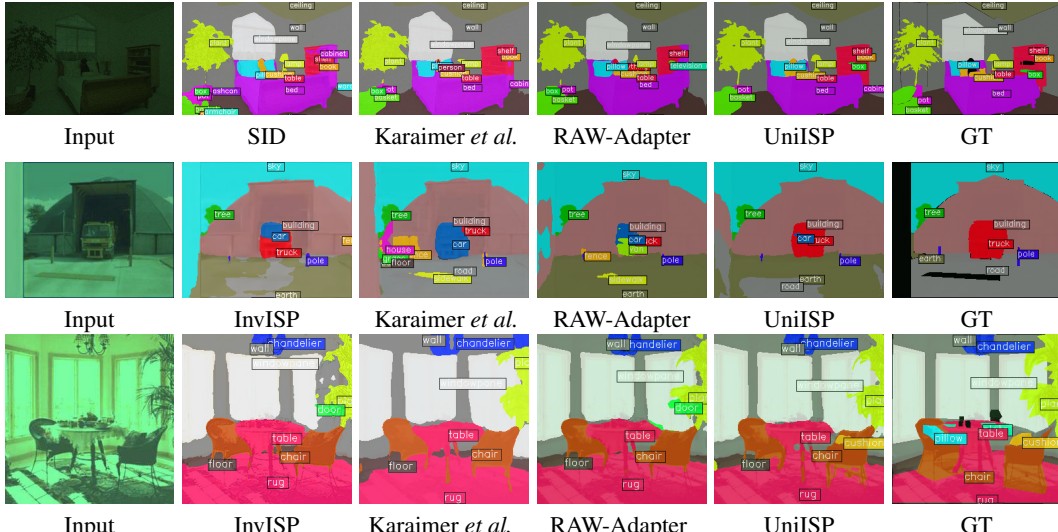

Figure 12: isualization of semantic segmentation results on ADE20K RAW. Three rows represent dark, normal and over-exposed scenarios respectively (from top to bottom).

inputs. Compared to existing methods such as RAW-Adapter and InvISP, UniISP demonstrates remarkable robustness and generalization, accurately delineating object boundaries and achieving close semantic alignment with the ground truth. Notably, UniISP preserves fine structural details and correctly identifies small and complex objects, even under severe lighting degradation. For instance, in low-light and over-exposure scenarios, UniISP maintains clear separation of categories such as "wall", "bed", "table", and "chair", and avoids ambiguous boundaries or missing objects commonly observed in other approaches. Overall, these qualitative results underscore the effectiveness of UniISP in semantic understanding from RAW images, setting a new benchmark for scene parsing under adverse visual conditions.

## E  LIMITATIONS

Despite the encouraging results, our work still has several limitations. First, the proposed multi-task framework requires paired RAW and RGB data for training. Although RAW data can be synthesized from RGB images, there are significant differences between synthesized RAW and genuine RAW data, both in terms of distribution and information content. Consequently, it remains challenging to acquire large-scale paired data, particularly for diverse real-world scenarios. This dependency on paired data may limit the scalability and generalization of UniISP in domains where such data is scarce or unavailable. Second, simultaneous optimization for both human perception and machine vision tasks increases training complexity, such as balancing multi-task objectives and resolving potential conflicts between perceptual and semantic requirements. Addressing these challenges may necessitate more sophisticated training strategies or adaptive loss functions. Future work could investigate semi-supervised or unsupervised approaches to reduce the reliance on paired datasets.

## F  ABLATION STUDY

The ablation study on the perceptual performance of the feature adapter for downstream tasks has been presented in Table 3 and Table 4. This section further validates the impact of the proposed modules on human visual perception on the ZRR dataset. As shown in Table 7 (a), a systematic ablation study was conducted on the proposed HAM module and the optical flow consistency mask. PSNR and SSIM were used as evaluation metrics to analyze the contribution of each component to image quality improvement. Here, "Base" refers to the U-Net-based baseline architecture, "MHA" denotes the standard multi-head attention mechanism, and "MCA" indicates the application of standard MHA in the channel dimension. The performance of MHA and MCA in the table represents the results

when replacing the HAM module with them, respectively. Table 7 (b) provides a detailed analysis of the impact of each component within the proposed HAM module, where RPE represents relative positional encoding, CA denotes channel attention, and SA refers to spatial attention.

Experimental results demonstrate that the proposed HAM module significantly outperforms traditional self-attention structures (MHA and MCA) in RAW-TO-RGB mapping tasks, with its effectiveness primarily stemming from the synergistic effect of channel attention (CA) and spatial attention (SA) mechanisms. Furthermore, the optical flow consistency mask mitigates alignment inaccuracies from occlusions and homogeneous regions by selectively applying loss to reliable areas, enhancing detail preservation and color fidelity in the output.

Table 7: Ablation study results.

(a) Performance comparison of different module combinations.

| Base | MHA | MCA | HAM | FlowMask | PSNR↑ | SSIM↑ |
|------|-----|-----|-----|----------|-------|-------|
| ✓ | | | | | 21.37 | 0.8312 |
| ✓ | ✓ | | | | 22.26 | 0.8421 |
| ✓ | | ✓ | | | 22.87 | 0.8486 |
| ✓ | | | ✓ | | 23.91 | 0.8584 |
| ✓ | | | ✓ | ✓ | **24.14** | **0.8614** |

(b) Performance analysis of the HAM module components.

| | HAM | | | Performance | |
|---|-----|----|----|-------------|-------|
| | RPE | CA | SA | PSNR↑ | SSIM↑ |
| MCA | | | | 22.87 | 0.8486 |
| MCA + RPE | ✓ | | | 23.02 | 0.8493 |
| MCA + CA | | ✓ | | 23.64 | 0.8556 |
| MCA + SA | | | ✓ | 23.23 | 0.8513 |
| MCA + RPE + CA | ✓ | ✓ | | 23.78 | 0.8562 |
| MCA + RPE + SA | ✓ | | ✓ | 23.35 | 0.8521 |
| MCA + CA + SA | | ✓ | ✓ | 23.85 | 0.8579 |
| **HAM (Ours)** | ✓ | ✓ | ✓ | **23.91** | **0.8584** |

# G   $\lambda$ STABILITY AND CONVERGENCE ANALYSIS

The EMA mechanism provides crucial stability by smoothing the weight transitions over time. Let $\lambda_{\text{instant}}^{(t)} = \mathcal{L}_{\text{machine}}^{(t)}/(\mathcal{L}_{\text{human}}^{(t)} + \mathcal{L}_{\text{machine}}^{(t)})$ be the instantaneous weighting using raw losses, and $\lambda_{\text{EMA}}^{(t)}$ be our proposed weighting using EMA-filtered losses as defined in Equation 7. The variance of our EMA-based $\lambda^{(t)}$ satisfies:

$$\text{Var}[\lambda_{\text{EMA}}^{(t)}] = \gamma^2 \cdot \text{Var}[\lambda_{\text{EMA}}^{(t-1)}] + (1-\gamma)^2 \cdot \text{Var}[\lambda_{\text{instant}}^{(t)}] \tag{10}$$

Since $0 < \gamma < 1$, we have:

$$\text{Var}[\lambda_{\text{EMA}}^{(t)}] \leq (1-\gamma)^2 \sum_{k=0}^{\infty} \gamma^{2k} \cdot \text{Var}[\lambda_{\text{instant}}^{(t-k)}] \leq \frac{1-\gamma}{1+\gamma} \cdot \text{Var}[\lambda_{\text{instant}}^{(t)}] \tag{11}$$

For $\gamma = 0.9$, this yields $\text{Var}[\lambda_{\text{EMA}}^{(t)}] \leq 0.053 \cdot \text{Var}[\lambda_{\text{instant}}^{(t)}]$, providing $\sim 20$x variance reduction.

The EMA-based $\lambda^{(t)}$ converges to an equilibrium point $\lambda^*$ that naturally balances human and machine vision objectives:

$$\lim_{t \to \infty} \lambda^{(t)} = \lambda^* = \frac{\mathbb{E}[\mathcal{L}_{\text{machine}}]}{\mathbb{E}[\mathcal{L}_{\text{human}}] + \mathbb{E}[\mathcal{L}_{\text{machine}}]} \tag{12}$$

The convergence follows from the contraction mapping principle:

$$|\lambda^{(t+1)} - \lambda^*| \leq \gamma|\lambda^{(t)} - \lambda^*| + (1-\gamma)|\varepsilon^{(t)}| \tag{13}$$

where $\varepsilon^{(t)}$ represents noise in loss estimates. Since $\gamma < 1$, the system converges exponentially.

To ensure robustness in extreme scenarios, we employ bounded adaptation:

$$\lambda_{\text{final}}^{(t)} = \text{clip}(\lambda_{\text{EMA}}^{(t)}, \lambda_{\min}, \lambda_{\max}) \tag{14}$$

Among them, $\lambda_{\min}$ and $\lambda_{\max}$ can be manually specified to define the control range. Furthermore, $\lambda$ supports dynamic adjustment during training.

