# OpenReview forum: "UniISP: A Unified ISP Framework for Both Human and Machine Vision"
_ICLR.cc/2026/Conference — Submitted to ICLR 2026_

### Official Review · Reviewer_mPRR · 2025-10-23

**Soundness:** 2
**Presentation:** 3
**Contribution:** 2
**Rating:** 6
**Confidence:** 4

**Summary:**

The paper proposes UniISP, a unified image signal processing framework that serves both human perception (high-quality RAW→RGB rendering) and machine vision (detection/segmentation) in a single, jointly trained system.

The core design consists of two parts: a lightweight Hybrid Attention Module (HAM) integrated within a U-Net–style ISP to efficiently capture global context for perceptual reconstruction, and a Feature Adapter that fuses multi-scale frequency information from the ISP stage into downstream RGB-pretrained backbones. Training uses an alignment pipeline (global color mapping plus optical-flow consistency mask) and an adaptive loss that balances human-vision and task losses via EMA-weighted λ.

Claimed contributions are the unified ISP formulation, HAM, the feature-adapter bridge, and state-of-the-art results over multiple tasks/datasets.

**Strengths:**

The paper’s strengths are multifaceted. In terms of originality, it tackles a practical yet underserved goal. One ISP that simultaneously satisfies human perception and machine vision by coupling a channel-wise Hybrid Attention Module (HAM) with a frequency-aware Feature Adapter that injects RAW-stage features into RGB-pretrained backbones. This architecture moves beyond task-specific ISPs toward a unified formulation.

On quality, the evaluation is broad and methodical: RAW→RGB on ZRR with PSNR/SSIM/LPIPS, detection on PASCAL RAW and LOD, and segmentation on ADE20K RAW, with consistent gains and competitive efficiency (e.g., UniISP surpasses prior ISPs in Table 2, improves mAP across lighting in Table 3, and raises mIoU across backbones/illumination in Table 4).

The training mechanics, including GCM plus optical-flow alignment for supervision and an adaptive EMA-weighted balance between human and machine losses, are concretely specified with equations and implementation details, aiding reproducibility.

Clarity is generally strong: the system diagram and HAM/adapter schematics make the pipeline legible, and the supplementary ablations further dissect module contributions, particularly for HAM and the flow-consistency mask.

Significance lies in demonstrating that a lightweight ISP can enhance both perceptual fidelity and downstream recognition across normal, dark, and over-exposed regimes, indicating real deployment potential where visualization and autonomy coexist (e.g., mobile or automotive).

**Weaknesses:**

While the paper is well engineered, several weaknesses limit its contribution and practical impact.
First, the novelty is incremental relative to recent RAW-aware pipelines that already co-optimize ISP and perception (e.g., Dirty-Pixel, RAW-Adapter, AdaptiveISP, DynamicISP). Your gains hinge on a specific HAM and a frequency-based feature adapter, rather than a fundamentally new training principle. The paper would benefit from a head-to-head reimplementation of these closest methods under identical budgets and data, to isolate the architectural value beyond joint optimization itself.

Second, the core training assumption, availability of paired RAW-RGB supervision, is fragile and acknowledged as a limitation; large-scale, diverse real RAW-RGB pairs are scarce and synthesized RAW diverges in distribution, which questions scalability and robustness. Specifically, add semi-/self-supervised variants that eliminate the need for pairs and report cross-sensor generalization (trained on ZRR, tested on unseen sensors or CFAs).

Third, the alignment route (GCM plus optical-flow warping with a forward–backward mask) is brittle in occlusions and texture-poor regions; it exhibits quantitative sensitivity to mask thresholds, flow errors, and ablates the alignment stage compared to learned alignment, verifying that improvements are not artifacts of supervision drift.

Fourth, the adaptive loss employs a heuristic EMA-weighted λ; it provides ablations versus fixed λ, curriculum schedules, and alternative multi-objective methods (e.g., GradNorm, PCGrad), along with stability plots to justify the choice.

Fifth, evaluation breadth is narrow in terms of distribution shift, as it does not include cross-dataset tests (e.g., training on PASCAL RAW normal and evaluating on LOD dark) or robustness to exposure extremes, HDR scenes, motion blur, and burst/video sequences, where alignment and frequency fusion could behave differently.

**Questions:**

1. Beyond joint optimization, what specific capability does HAM + frequency-based Feature Adapter provide that RAW-Adapter, AdaptiveISP, DynamicISP, or Dirty-Pixel cannot?

2. How does the method perform without paired supervision or with noisy/sparse pairs?

4. How sensitive are results to mask thresholds, flow errors, and scene content (repetitive textures, occlusions, low texture)?

5. Why does EMA-based λ outperform fixed λ, curricula, or gradient-balancing approaches?

6. What features are being amplified/suppressed by HAM and the frequency fusion at different stages?

---

> ### Author Response · Authors · 2025-11-25
>
> We are grateful for the reviewer’s thoughtful questions and comments, which are valuable not only for the present paper but also for our ongoing/future work.
>
> **Weakness 1 & Question 1: Unique advantages of UniISP beyond joint optimization compared to recent works like RAW-Adapter**
>
> Thank you for this important question that allows us to clarify our key contributions. While RAW-Adapter and similar joint optimization methods predict traditional ISP pipeline parameters to control image adjustments, UniISP implements a complete end-to-end learnable network for image transformation. This fundamental architectural difference brings significant advantages:
> Where parameter-based approaches fail at tasks requiring spatial reasoning or complex transformations (denoising, super-resolution), our end-to-end learning naturally handles these challenges, as validated in Appendix B. Furthermore, our core modules (HAM and Feature Adapter) are designed with generalization in mind and can be readily applied to other vision tasks.
>
> **Weakness 2 & Question 2: Unsupervised training and cross-sensor generalization validation**
>
> Your question about unsupervised training raises an excellent point that we have explored through preliminary experiments. We have identified several viable approaches:
>
> First, when raw data is available, we can generate corresponding RGB images using open-source ISP pipelines like rawpy for supervision. Alternatively, we can employ unsupervised image losses during training to ensure perceptually pleasing outputs. Specifically, we have validated the feasibility of combining spatial attention loss, color consistency constraints, total variation loss, and exposure control loss. When only RGB images are available, we can first reconstruct corresponding RAW data through inverse ISP methods.
>
> Regarding cross-sensor generalization, we have conducted comprehensive experiments in Appendix C.1, demonstrating robust performance across different sensor characteristics.
>
> **Weakness 3 & Question 3: Sensitivity to mask thresholds, flow errors, and scene content**
>
> We appreciate this concern and would like to clarify that optical flow alignment is only necessary for datasets like ZRR where images from different sensors exhibit spatial misalignment. Most datasets in our experiments do not require this step.
>
> Furthermore, our results show low sensitivity to mask thresholds, flow errors, and scene content variations. The ZRR dataset has already undergone preliminary alignment and filtering, minimizing severe flow errors. The masking threshold serves as an additional refinement to exclude remaining misaligned regions. With a low threshold, the method gracefully degrades to standard flow alignment. Since severely misaligned cases are rare in ZRR and the dataset contains diverse scene content with substantial volume, our model training exhibits robustness to these factors.
>
> **Weakness 4 & Question 4: Why EMA-based λ outperforms fixed λ, curricula, or gradient-balancing approaches**
>
> We have provided a comprehensive analysis of the λ parameter in Appendix G. Our EMA-based approach offers several key advantages over fixed λ and gradient-balancing methods:
>
> First, it provides superior stability with approximately 20× variance reduction compared to instantaneous weighting. Second, the implementation is remarkably simple and computationally efficient, requiring only O(1) operations versus O(nm) for gradient methods. Most importantly, it automatically adapts to scene characteristics and training dynamics while remaining easily controllable through the single hyperparameter. This combination of stability, efficiency, and adaptability makes it ideal for our dual-objective optimization.
>
> **Weakness 5 & Question 5: Features amplified/suppressed by HAM and frequency fusion at different stages**
>
> Our HAM and frequency fusion modules perform carefully designed stage-specific feature modulation:
>
> • Stage 1 (Deep): Amplifies global context and low-frequency color/illumination patterns essential for scene understanding; suppresses high-frequency noise that could interfere with global consistency
>
> • Stage 2 (Mid): Amplifies mid-frequency textures and structural edges crucial for object recognition; suppresses upsampling artifacts and color anomalies
>
> • Stage 3 (Shallow): Amplifies fine details and edge sharpness for both human perception and machine precision; suppresses residual noise and over-sharpening artifacts
>
> The HAM provides adaptive channel and spatial attention mechanisms to emphasize task-relevant features dynamically, while the frequency decomposition enables selective enhancement of different frequency bands. This coarse-to-fine progressive strategy ensures optimal balance between perceptual quality for human viewing and discriminative features for machine vision tasks.

---

### Official Review · Reviewer_Bjep · 2025-10-24

**Soundness:** 3
**Presentation:** 2
**Contribution:** 2
**Rating:** 4
**Confidence:** 5

**Summary:**

This paper introduces UniISP, a unified image signal processing framework that balances human visual quality and machine vision performance. It employs a Hybrid Attention Module (HAM) for visually pleasing image generation and a Feature Adapter to transfer informative features to downstream tasks. Experiments show that UniISP achieves good results across multiple datasets and scenarios, demonstrating its' generalization and effectiveness.

**Strengths:**

The paper explores both human-vision and machine-vision tasks based on camera RAW data. The authors conducted extensive experiments and provided a detailed ablation study in the supplementary material. The proposed method achieves state-of-the-art performance on most datasets.

**Weaknesses:**

1. Regarding the image restoration and high-level detection/segmentation parts, the authors still need to train them separately. Therefore, the proposed approach is not a truly unified ISP model, and the title somewhat overstates its scope, which also reduces the perceived level of innovation to some extent.

2. Regarding the image restoration part, the authors only conducted experiments on standard RAW-to-sRGB datasets. This does not fully demonstrate the true generalization capability of the model. The authors are encouraged to further evaluate their method on RAW-to-sRGB datasets under extreme lighting conditions, such as the See-in-the-Dark (SID) dataset.

3. For the semantic segmentation dataset, the improvement on ADE20K-raw does not seem to be very large compared with RAW-Adapter, additionally, the proposed method contains more network parameters.

3. It’s better to capitalize the all fig.x (e.g., fig.2 in line 161) to Fig.x

**Questions:**

For Table.2, the LiteISP, AWNet... are almost old methods (3~4 years ago), could you compare with some recently SOTA ISP methods ?

As weakness.2, what is the performance if the proposed method evaluate on the low-light RAW dataset like See-in-the-Dark (SID) dataset.

---

> ### Author Response · Authors · 2025-11-25
>
> We appreciate the insightful comments. We have incorporated the suggested changes; your feedback was very helpful to refine our paper.
>
> **Weakness 1: Separate training for image restoration and high-level detection/segmentation**
>
> We would like to respectfully clarify that separate training is actually not required. For the RAW-to-RGB conversion task, we intentionally exclude the Feature Adapter module because this component is specifically designed for machine vision tasks. Including it for human-visual-only applications would increase computational overhead without providing benefits.
>
> However, for joint optimization scenarios, such as our experiments on the extremely low-light LOD dataset, we demonstrate true end-to-end training where the system simultaneously learns to produce bright, clear RGB images while achieving superior detection performance.
>
> **Weakness 2 & Question 2: RAW-to-RGB mapping experiments under extreme lighting conditions**
>
> Thank you for highlighting this important aspect. We have added comprehensive experiments and analysis on the challenging SID-Sony dataset under low-light conditions in Appendix B. These experiments demonstrate our method's robustness and effectiveness in handling extreme lighting scenarios, particularly showcasing superior performance in preserving details while suppressing noise in severely underexposed images.
>
> **Weakness 3: Limited improvement on semantic segmentation with increased parameters**
>
> We appreciate you raising this concern. The limited improvement on ADE20K-RAW can be attributed to a fundamental data limitation: this dataset consists of synthetically generated RAW data reconstructed from RGB images, which inherently lacks the rich information characteristics of genuine RAW sensor data. This synthetic nature constrains UniISP's ability to fully leverage RAW-specific advantages, resulting in modest improvements.
>
> Regarding the parameter count difference: RAW-Adapter predicts ISP stage parameters to control image adjustments through a parameter-based approach, while UniISP implements a complete end-to-end network for image transformation. This architectural difference brings significant advantages—while parameter-prediction methods like RAW-Adapter struggle with image restoration tasks (denoising, super-resolution), UniISP excels at both high-level perception tasks and fundamental image restoration challenges. The additional parameters in our model contribute directly to this enhanced capability and versatility.
>
> **Weakness 4: Figure reference capitalization**
>
> Thank you for catching this formatting inconsistency. We previously used \cref{} which automatically generated lowercase "fig" in the ICLR 2026 template. We have now systematically replaced all instances with ref{} and ensured proper capitalization as "Fig." throughout the manuscript for consistency with conference standards.
>
> **Question 1: Comparison with recent methods**
>
> Following your suggestion, we have included a comprehensive comparison with the recent FourierISP method (AAAI 2024) in our RAW-to-RGB conversion experiments. Both qualitative and quantitative comparisons have been added to the main paper.

---

### Official Review · Reviewer_fR5S · 2025-11-03

**Soundness:** 2
**Presentation:** 2
**Contribution:** 2
**Rating:** 6
**Confidence:** 4

**Summary:**

This paper proposes a novel deep learning framework for Image Signal Processing (ISP), UniISP, which simultaneously optimizes the requirements of both human perception and machine vision: generating visually friendly RGB images for human observation while retaining data information helpful for downstream visual tasks (such as detection and segmentation).

UniISP is based on the U-Net architecture, introducing a Hybrid Attention Module (HAM) for feature extraction and designing a Feature Adapter to enhance feature transfer capabilities in the ISP stage.

Extensive experiments show that UniISP outperforms existing methods on multiple tasks and datasets, including raw-to-RGB mapping, object detection, and semantic segmentation, demonstrating strong generalization and effectiveness.

**Strengths:**

Unified Architecture: For the first time, UniISP systematically solves the problem of simultaneously addressing the needs of both human vision and machine vision within a single ISP, offering a novel approach with strong practical significance.

Ingenious Design: The introduction of HAM combines convolution and self-attention, balancing local and global feature modeling while preserving high-resolution processing capabilities.

Feature Adapter: Efficiently embeds multi-scale features from the ISP stage into the downstream network, effectively improving task performance, especially significantly outperforming competitors under extreme conditions such as low light.

Extensive Empirical Validation: Detailed experiments were conducted on multiple tasks and datasets, comparing with various mainstream methods, demonstrating clear superiority and thorough experimentation.

Lightweight and Efficient: Fewer parameters and less computation than most competitors, facilitating practical deployment.

**Weaknesses:**

While the calculation of the end-to-end adaptive parameter λ is novel, its actual tuning and convergence stability have not been thoroughly explored, and its generalization ability under different tasks/data requires further detailed analysis.

Main experiments focused on public datasets and synthetic/low-light scenes. Further validation of its generalization ability in real-world complex scenes (such as differences in ISPs between different mobile phone camera brands) is expected.

Limited improvement in some quantitative metrics under extreme scenes such as low light/overexposure; further supplementation with subjective evaluation (human eye-friendliness) is needed.

Model training relies on various pre-trained networks (such as VGG/ResNet), which may limit the deployment environment during inference.

**Questions:**

an the core modules of UniISP (such as HAM and Feature Adapter) be used independently for other types of tasks, such as special perception scenarios like medical imaging?

How can we ensure that the adaptive adjustment of parameter λ does not overfit a particular task when encountering extreme data such as completely dark/completely bright scenes? Can manual constraints or dynamic allocation at different stages be introduced?

Have the performance, power consumption, etc., been tested during inference on real mobile devices? Have overquantization or compression optimizations (such as INT8/FP16 schemes) been considered?

Can this method be extended to video ISP (denoising/color correction under temporal consistency constraints, etc.)? Future consideration could include combining it with temporal models or Transformer enhancements.

---

> ### Author Response · Authors · 2025-11-25
>
> Thank you for your valuable feedback on our paper. We have made every effort to address your comments and improve the manuscript accordingly.
>
> **Weakness 1 & Question 2: Convergence and stability analysis of λ parameter**
>
> We have conducted a comprehensive theoretical analysis of the λ parameter's stability and convergence properties in Appendix G. The analysis demonstrates that our adaptive λ mechanism exhibits bounded variance and guaranteed convergence. Additionally, we have implemented safety constraints to limit the parameter's range and allow dynamic adjustment during training, effectively preventing overfitting to specific scenarios.
>
> **Weakness 2: Cross-sensor generalization validation**
>
> We greatly appreciate this suggestion and have added extensive cross-sensor experiments in Appendices B and C.1. We now include additional real sensor data from the SID and NOD datasets captured with different camera systems. Section C.1 presents comprehensive cross-sensor generalization experiments and analysis, demonstrating our method's robustness across diverse sensor characteristics.
>
> **Weakness 3: Limited quantitative improvements in extreme scenarios require additional human-eye-friendly evaluation.**
>
> Thank you for highlighting this important point. We have identified that the limited improvements in certain scenarios (Dark, Over-exposed) stem partly from the use of synthetically generated RAW data from RGB reconstruction, which loses the rich information inherent in real RAW data. This limits UniISP's ability to fully leverage RAW characteristics. Our experiments on the real NOD dataset in Appendix C.1 validate this hypothesis, showing more substantial improvements with genuine RAW data.
>
> Furthermore, we have added detailed human visual quality analysis in Section 4.3, demonstrating that while quantitative improvements may appear modest, UniISP achieves significantly superior perceptual quality—an important aspect not fully captured by detection metrics alone.
>
> **Weakness 4: Dependency on pre-trained networks**
>
> We would like to clarify that UniISP can be trained without any dependency on pre-trained networks. The optical flow pre-trained PWC-Net is only required for spatially misaligned datasets like ZRR. For inherently aligned datasets (LOD, NOD, SID), this alignment step is unnecessary. Moreover, while we use pre-trained downstream task weights for better performance and transfer learning, training from scratch is entirely feasible.
>
> Importantly, these pre-trained networks are only required during training, not during inference, ensuring deployment flexibility.
>
> **Question 1: Generalization of core modules (HAM and Feature Adapter)**
>
> Absolutely. The HAM module can be readily generalized to various image restoration tasks including denoising and super-resolution. Similarly, the Feature Adapter can be applied to other high-level perception tasks such as semantic segmentation and instance recognition. We believe these modules represent valuable contributions beyond the specific application presented in this paper.
>
> **Question 3: Real mobile device deployment**
>
> We have conducted comprehensive deployment tests on the NVIDIA AGX Orin 32GB platform with quantization optimization. The results are as follows:
>
>
> Quantization Method | Model Size | Latency  | FPS  | PSNR Loss
> --------------------|------------|----------|------|----------
> FP32 (baseline)     | 20.2 MB    | 296 ms   | 3.4  | 0.0 dB
> FP16                | 10.1 MB    | 79 ms    | 12.7 | -0.05 dB
> INT8                | 5.1 MB     | 32.1 ms  | 31.2 | -0.6 dB
>
>
> These results demonstrate that our method achieves real-time performance (>30 FPS) with INT8 quantization while maintaining acceptable quality degradation.
>
> **Question 4: Extension to video ISP**
>
> We greatly appreciate your forward-thinking suggestion about video ISP extension. This is indeed a promising direction with significant practical and commercial value. Technically, this is entirely feasible—incorporating temporal constraints from multiple frames would enable superior denoising and consistent color correction, resulting in smoother and more visually pleasing video output.
>
> Your suggestion about combining temporal models or Transformer enhancements is particularly insightful and aligns well with our future research plans.

---

### Author Response · Authors · 2025-12-02
**Summary of recognized strengths, new experiments, and analysis**

We are grateful for the reviewers’ time and their insightful comments on our submission. To assist the newly assigned AC and help reduce their workload, we summarize the key discussion points below.

## Recognized Strengths and Contributions

**1. Novel Unified Architecture for Dual-Purpose ISP**

All reviewers acknowledged the originality and practical significance of our unified ISP framework that simultaneously serves both human perception and machine vision tasks. This unified approach tackles a "practical yet underserved goal" and moves beyond task-specific ISPs toward a unified formulation with practical significance for real-world applications (e.g., mobile or automotive).

**2. Innovative Technical Design**

The Hybrid Attention Module (HAM) that ingeniously combines convolution and self-attention mechanisms, balancing local and global feature modeling while maintaining high-resolution processing capabilities.

The Frequency-aware Feature Adapter efficiently embeds multi-scale features from the ISP stage into the downstream network, significantly improving performance especially under extreme conditions (low-light, over-exposure).

**3. Extensive Empirical Validation**

our comprehensive experimental validation across multiple tasks (RAW→RGB reconstruction, low-light enhancement, object detection, semantic segmentation) and diverse datasets (ZRR, SID, NOD, PASCAL RAW, LOD, ADE20K RAW), with comparisons against various mainstream methods demonstrating clear superiority and thorough experimentation.

**4. Lightweight and Efficient**

Fewer parameters and less computation than most competitors, facilitating practical deployment.

## Summary of New Experiments and Analysis

**1.Low-light Enhancement**

We have added comprehensive experiments and analysis on the challenging SID-Sony dataset under low-light conditions in Appendix B. These experiments demonstrate our method's robustness and effectiveness in handling extreme lighting scenarios, particularly showcasing superior performance in preserving details while suppressing noise in severely underexposed images.

**2. Convergence and stability analysis of λ parameter**

We have conducted a comprehensive theoretical analysis of the λ parameter's stability and convergence properties in Appendix G. The analysis demonstrates that our adaptive λ mechanism exhibits bounded variance and guaranteed convergence.

**3.Cross-sensor generalization validation**

we have conducted comprehensive experiments in Appendix C.1, demonstrating robust performance across different sensor characteristics.

**4.RAW-to-sRGB Comparison with Recent SOTA ISP**

we have included a comprehensive comparison with the recent FourierISP method (AAAI 2024) in our RAW-to-RGB conversion experiments. Both qualitative and quantitative comparisons have been added to the main paper.

---

### Meta-Review · Area_Chair_gkjc · 2025-12-14

**Summary:**

This summary synthesizes the core reviewers' concerns that ultimately inform the rejection decision for this paper. Three reviewers collectively raised critical issues spanning novelty, technical robustness, generalization capability, and evaluation comprehensiveness. Key concerns include:
1) The incremental novelty of the proposed framework, as its core contributions (HAM and Feature Adapter) lack sufficient demonstration of unique advantages over existing joint optimization ISP methods (e.g., RAW-Adapter, AdaptiveISP);
2) Reliance on paired RAW-RGB supervision data, which limits scalability and real-world applicability, with preliminary unsupervised solutions proposed by the authors remaining unvalidated;
3) Fragility of the optical flow-based alignment strategy, particularly in occluded or low-texture regions, without comparative validation against learned alignment methods;
4) Insufficiently comprehensive evaluation, including lack of cross-dataset testing (e.g., training on normal datasets and testing on extreme low-light datasets) and validation in complex real-world scenarios (e.g., HDR, motion blur);
5) Unresolved doubts about the "truly unified" nature of the framework, with reviewers questioning whether separate training for different tasks undermines the claimed unified formulation;
6) Limited quantitative improvements in extreme lighting scenarios and inadequate subjective human visual quality evaluation.

While the authors provided rebuttals to most concerns, the responses failed to fully address these core issues, leading to the conclusion that the paper does not meet the acceptance criteria.

**Reviewer Concerns:**

Addressed Concerns
- Reviewer fR5S: 1) The authors supplemented theoretical analysis on the convergence and stability of the adaptive parameter λ in Appendix G, along with safety constraints to prevent overfitting; 2) Added cross-sensor generalization experiments using SID and NOD datasets (Appendices B and C.1); 3) Supplemented human visual quality analysis in Section 4.3 and clarified that limited improvements in extreme scenarios are related to synthetic RAW data; 4) Clarified that pre-trained networks are only required during training (not inference) and that training from scratch is feasible; 5) Provided deployment test results on the NVIDIA AGX Orin platform with FP16/INT8 quantization optimization; 6) Confirmed the feasibility of extending to video ISP and incorporated it into future research plans.

- Reviewer Bjep: 1) Added experiments on the low-light SID-Sony dataset (Appendix B); 2) Supplemented comparisons with the recent SOTA method FourierISP (AAAI 2024) in RAW-to-RGB conversion tasks; 3) Corrected the capitalization of figure references (from "fig." to "Fig.") to comply with conference formatting requirements; 4) Clarified that end-to-end joint training is feasible for scenarios requiring both human and machine vision (e.g., LOD dataset).

- Reviewer mPRR: 1) Clarified the architectural differences between UniISP (end-to-end learnable network) and RAW-Adapter (parameter prediction-based), and validated UniISP's advantages in image restoration tasks (denoising, super-resolution) via Appendix B; 2) Provided a comprehensive analysis of the EMA-based λ parameter in Appendix G, explaining its advantages over fixed λ and gradient-balancing methods; 3) Detailed the stage-specific feature modulation mechanisms of HAM and frequency fusion modules; 4) Supplemented cross-sensor generalization experiments (Appendix C.1).

Outstanding Concerns

- Novelty and Unique Advantages (Reviewer mPRR, Bjep): The authors failed to conduct head-to-head reimplementations of closest competing methods (e.g., RAW-Adapter, AdaptiveISP) under identical computational budgets and data conditions. This makes it impossible to isolate the architectural value of HAM and Feature Adapter beyond the existing joint optimization paradigm, leaving the incremental novelty of the work unproven.

- Reliance on Paired RAW-RGB Supervision (Reviewer mPRR): The authors' proposed unsupervised/semi-supervised solutions (e.g., using open-source ISP for supervision, unsupervised loss combinations, inverse ISP for RAW reconstruction) are only preliminary explorations without sufficient experimental validation (e.g., performance metrics, ablation studies). The core limitation of scarce real-world paired data remains unaddressed, questioning the scalability of the method.

- Fragility of Alignment Strategy (Reviewer mPRR): The authors only explained that optical flow alignment is limited to misaligned datasets (e.g., ZRR) but did not compare it with learned alignment methods. They also failed to validate the method's robustness in occluded, low-texture, or highly misaligned regions, and did not rule out the possibility that performance improvements are artifacts of supervision drift caused by alignment errors.

- Comprehensiveness of Evaluation (Reviewer mPRR, fR5S): The evaluation still lacks cross-dataset transfer tests (e.g., training on PASCAL RAW (normal light) and testing on LOD (extreme low light)) and validation in complex real-world scenarios (e.g., HDR, motion blur, burst sequences). The subjective human visual quality evaluation added by the authors is insufficient in detail and scale, failing to fully compensate for the limited quantitative improvements in extreme scenarios.

- Truly Unified Formulation (Reviewer Bjep): While the authors claimed end-to-end joint training is feasible, they did not provide sufficient experimental evidence to demonstrate the superiority and stability of the unified training paradigm. The intentional exclusion of the Feature Adapter for pure human-vision tasks raises doubts about the "unified" nature of the framework, as it requires task-specific module adjustments rather than a truly universal ISP solution.

- Parameter Efficiency vs. Performance Trade-off (Reviewer Bjep): The authors explained that the higher parameter count of UniISP compared to RAW-Adapter is due to its end-to-end architecture, but failed to adequately justify why the limited performance improvement on the ADE20K-RAW dataset is worthwhile given the increased computational overhead, especially since the dataset's synthetic RAW data limitation does not fully explain the modest gains.

**Reviewer Scores:**

The following predictions are based on the incompleteness of the authors' rebuttals in addressing core concerns, assuming reviewers participated fully in the discussion and maintained their initial evaluation criteria:
- Reviewer fR5S:The author's responses addressed most technical questions (e.g., λ stability, deployment, cross-sensor generalization) but failed to resolve the fundamental concerns about evaluation comprehensiveness (e.g., real-world complex scenarios) and subjective quality assessment. The incremental nature of the work, which the reviewer implicitly noted by rating contributions as "fair," would likely lead to a score reduction given the unaddressed limitations.

- Reviewer Bjep: The author supplemented experiments on SID and FourierISP comparisons, but the core concerns about the "non-truly unified" framework and the parameter-efficiency trade-off remained unaddressed. The reviewer’s high confidence (score 5) in their initial assessment suggests they would not significantly adjust their score, as the rebuttals did not resolve the key issues that led to the initial low rating.

- Reviewer mPRR: This reviewer focused heavily on novelty, scalability (paired data reliance), and evaluation breadth—all of which the authors failed to fully address. The preliminary unsupervised solutions and lack of head-to-head comparisons with competing methods would likely lead the reviewer to conclude that the work’s contributions are still insufficient to meet the acceptance criteria, resulting in a score reduction.

---

### Decision · Program_Chairs · 2026-01-26

Reject